# Charting Cellular Networks in the Tumor Microenvironment: Graph Visualizations in Highly-Multiplexed Breast Cancer Tissue

Lara Elvevåg[1], Youssef Wally[*1], Benjamin Ricaud[1], Vladan Milosevic[2], and Elisabeth Wetzer[1]

[1]Department of Physics & Technology, UiT The Arctic University of Norway, Tromsø, Norway
[2]Department of Clinical Medicine, University of Bergen, Bergen, Norway
{lara.m.elvevag, youssef.m.wally, benjamin.ricaud, elisabeth.wetzer}@uit.no
{v.milosevic}@uib.no

## Abstract

The tumor microenvironment, composed of diverse interacting cells, plays a pivotal role in cancer progression by influencing tumor growth, immune evasion, and therapeutic outcomes. In this ongoing work, we propose the use of interactive graph visualizations as a versatile tool for exploring and analyzing biological data to unravel cell communication patterns in the tumor microenvironment and identify graph representations of the multiplex data, which can subsequently be leveraged by graph neural networks to contextualize cell interactions with patient outcomes.

## 1   Introduction

Developments in immunotherapy have transformed cancer treatment by activating a patient's immune system to recognize and target cancer cells [1–3]. However, the variability in patient responses has driven interest in understanding the underlying mechanism of these therapies more deeply. The tumor microenvironment (TME) plays a critical role in cancer progression, as it consists of a number of different cells which influence the tumor growth and immune response, by e.g. allowing cancer cells to evade immune detection or regulating their proliferation rate [4–6]. To improve therapeutic outcomes and identify cancer biomarkers that can be used for early cancer detection, a better understanding of the TME and its how cells communicate with each other is essential.

Developments in spatial omics and imaging technologies enable spatial profiling of gene and protein expression in tissue which allows for an unprecedented resolution of cells in terms of functionality [7]. With the ability to measure a large number of expressed proteins or genes per cell in the TME, the complexity of this data grows rapidly, making manual data inspection very challenging. Therefore, computational tools for analyzing this highly complex data, as well as visualization tools for data exploration and result interpretations are required.

In this ongoing work, we propose to use interactive graph visualization tools such as Gephi to qualitatively assess cell classification approaches which lack annotated ground truth labels and visualize cell communities in the TME to identify patterns of cell-cell interaction. Which can then be leveraged not only for visualization, but also for training Graph Neural Network (GNN), enabling the model to learn relationships in the graph. This integration bridges visualization with graph-based learning for more interpretable downstream analysis.

## 2   Related Work

Recent studies targeted to study cell interactions in tissue have exploited graph visualization tools. Karimi et al. used graph visualization tools for protein-protein interaction analysis in pancreatic Ductal adenocarcinoma [8]. So-called topological tumor graphs were derived from H&E stained whole slide images and combined with omic data to analyze melanoma histology in [9]. Cellular graphs were also proposed by Wang et al. to model the TME, alongside population graphs, capturing inter-patient similarities given their respective cellular graphs were proposed to study patterns in breast tumor microenvironments [10]. Protein interactions and networks were studied using graphs within the tumor in prostate cancer [11] and Rohail et al. proposed graph theoretical concepts to understand the TME of hematolymphoid cancer in H&E stained histological images [12].

## 3   Data

We analyze imaging mass cytometry (IMC) data, which detects up to 50 protein markers at subcellular resolution using metal-tagged antibodies and time-of-flight mass spectrometry. This dataset, derived from stained breast cancer tissue sections, provides spatially resolved, high-dimensional single-cell data to capture the phenotypic diversity of the TME.

---

[*]Corresponding Author.

# 4 Methodology

Spatial analysis of IMC data typically involves cell segmentation, using tools like Mesmer [13] for nucleus and cell segmentation, followed by cell classification to identify recurring spatial patterns. Cell classification remains challenging due to staining variability, imaging differences, and limited annotated ground truth data. Examples of such methods are MAPS [14] and ASTIR [15], however different maker panels targeted to study particular cell (sub-)types are not necessarily captured well or at all by existing methods.

We apply a weighted Gaussian mixture model and logical rules based on expected biomarker combinations to achieve hierarchical cell type annotations across four levels of granularity in breast cancer tissue microarray images, with a focus on immune cells and fibroblasts.

## 4.1 Graph Construction

We argue that interactive graph visualizations can be useful in various ways.

**I.) Classification Assessment**
A bipartite graph can evaluate classification quality by linking cell labels to expressed biomarkers, with edges indicating biomarker expression in specific cell types (figure 2). This visualization intuitively reveals which biomarkers are associated with each cell type, enabling an easily interpretable quality check to ensure alignment with biological patterns and highlighting potential classification issues.

**II.) Cell Niche Detection**
Cell niches can be visualized as a graph where nodes represent cell types and edges indicate neighboring cell types, capturing cell communities and their interactions. This approach helps identify significant communities in specific subpopulations or patient groups for further statistical analysis.

**III.) Protein Expression in Cell Niche Environments**
By combining these approaches, we link cell-biomarker bipartite graphs with cell neighborhood data to map biomarkers to entire cell communities. This reveals unique biomarker expression patterns within communities and their biological significance.

**IV.) Cell Subtype Discovery** A graph of biomarkers, with edges indicating co-expression and thickness reflecting frequency, can reveal patterns that define cell subtypes (figure 1). Analyzing strongly connected clusters in this graph helps identify distinct cell subtypes.

# 5 Implementation

The graphs were made using the Python library networkx [16] and the open-source software gephi

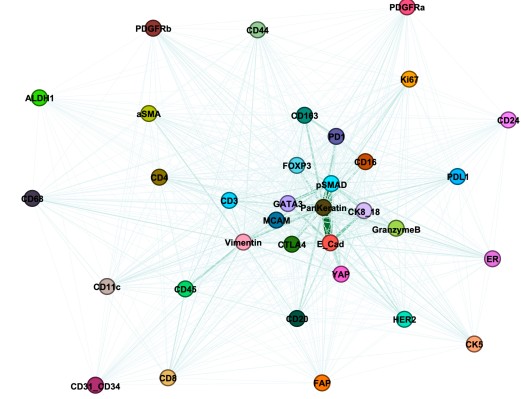

**Figure 1.** Co-occurrence graph of biomarkers. Edge thickness is proportional to co-occurrence frequency.

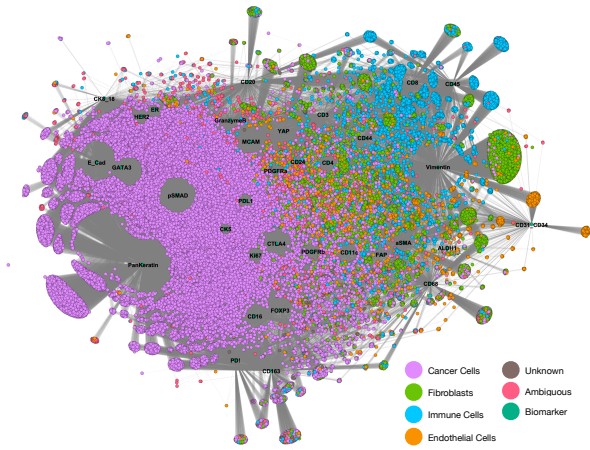

**Figure 2.** Bipartite graph of nodes; which represent either cells or biomarkers, with edges indicating the presence of a given biomarker within a cell.

[17]. The graphs are displayed following a *force atlas* layout, keeping the connected nodes closer and pushing away unconnected ones, highlighting existing clusters and communities in the graph. Defining the graphs with networkx enable the direct use of GNN implementations (such as Pytorch-Geometric), streamlining further analysis.

Gephi makes it easy to share visualizations through web pages or Gephi files, requires no Python expertise, and supports interactive exploration, making it ideal for interdisciplinary groups.

# 6 Conclusions

Interactive graph visualizations simplify complex biological data, revealing cell interactions and biomarker patterns in the tumor microenvironment. Combined with tools like Gephi and graph neural networks, they enable deeper insights and connections to patient outcomes.

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
