# OpenReview forum: "Charting Cellular Networks in the Tumor Microenvironment: Graph Visualizations in Highly-Multiplexed Breast Cancer Tissue"
_NLDL.org/2026/Abstracts_Track — NLDL 2026 Abstracts_

### Official Review · Reviewer_afDf · 2025-10-25

**Soundness:** 2
**Correctness:** 2
**Rating:** 4
**Confidence:** 3

**Summary:**

The abstract proposes novel applications of GNNs based on stained breast cancer tissue. The objective of the work is to capture the phenotype diversity. The authors construct graphs to support classification, cell niche detection, protein expression, and subtype discovery.

**Strengths:**

The paper sketches a novel line of inquiry into the medical domain which is on theme for NLDL. It provides a direction on how to make more meaningful data visualizations and attempts to unify exploratory visualization and computational modeling

**Weaknesses:**

# Research Objectives
It would be useful if there were discrete research questions in the introduction. When trying to determine which specific questions or goals, the data section seemed to be the most clear.

The section on graph construction lists four conceptual uses (classification assessment, niche detection, etc.), but lacks mathematical definitions, graph structure details (e.g., how edges are weighted, thresholds for co-expression), and examples of output metrics.

# Reproducibility
The authors should consider Gephi's configuration. Gephi’s  ForceAtlas2 layout  is stochastic: node positions depend on random initialization and iterative physical simulation. Unless the random seed and layout parameters (scaling, gravity, iterations, etc.) are explicitly fixed and reported, two users running the same data can get different-looking graphs. This makes it visually intuitive but not deterministic or reproducible.

# Typos/Grammar
1. ..the TME and <<its how>> cells communicate with each other is essential.
2.  ...in tissue <<,>> which allows for an unprecedented resolution...
3. ...as well as visualization tools for data exploration and result interpretation<<s>>, are required.
4. ... can then be leveraged not only for visualization but also for training <<a Graph Neural Network (GNN)>>, enabling the model to learn relationships in the graph.

---

### Official Review · Reviewer_BdFx · 2025-10-28

**Soundness:** 3
**Correctness:** 3
**Rating:** 4
**Confidence:** 3

**Summary:**

In this abstract, authors propose the use of interactive graph visualizations in the case of tumor microenvironment which is essential in cancer detection. Authors in 4.1 list some possible outcomes of the proposed strategy.

**Strengths:**

The abstract is well motivated and it addresses a fundamental challenge in early cancer detection. The authors are aware of the recent and relevant literature (Section 2). The dataset employed consists of up 50 protein markers, ensuring a detailed profiling.

**Weaknesses:**

It is important to emphasize that this paper is far from my field of expertise, so my assessment may not be entirely accurate. However, for the purposes of this presentation, I would like to point out a few details that could be improved:
- W1: It is not clear how this work differs from previous work in section 2.
- W2: Figure 2 is cited before Figure 1.
- W3: Although the authors list five possibilities in section 4.1, only two are related to the figures in the paper, and there is no practical mention of what types of insights can be derived. The section remains somewhat vague, and it would be useful to provide at least one concrete, graphic example to support it.
- W4: The first part of Section 4 is more accurately described as related work, and the actual methodology (lines 092-097) remains vague.

I encourage the authors to review some of these details to ensure better dialogue with the community.

---

### Official Review · Reviewer_k8er · 2025-10-31

**Soundness:** 3
**Correctness:** 3
**Rating:** 4
**Confidence:** 3

**Summary:**

The authors propose to use interactive graph visualization tools to qualitatively assess cell classification algorithms. Furthermore, they intend to use graph visualization tools to visualize cell communities in the TME, which plays an important role in the development of cancer cell growth rate, in order to discover patterns of cell-cell interactions.

The authors present several graph construction approaches for visualizing cell subtypes, cell communities and cell-cell interactions. The authors suggest that graph neural networks (GNNs) may be used in conjunction with their graph construction approach to relate cell interactions to patient outcomes.

**Strengths:**

- The abstract is clearly written, with vivid graph illustrations.
- The idea to use graph visualization tools for identifying biomarkers and cell-cell interactions is technically sound, and fits well with the data domain.
- The authors propose several graph construction approaches for analysis of cell types and interactions, including approaches for identifying cell biomarkers, cell subtypes, cellular communities and cell-cell interactions.
- The authors claim that their method can be effectively utilized by GNNs to model graph relationships, providing a promising future research direction.

**Weaknesses:**

- The related work section does not frame the research contribution well. The authors state that the use of graph visualization approaches has already been exploited in this data domain, yet they do not explain what makes their approach novel/better compared to these.
- The primary research contribution is unclear. Is it the use of graph visualization tools in the data domain? Is it the data analysis and methodology? Is it the method of graph construction itself?
- The authors should reflect on how they intend to approach modelling graph relationships using GNNs.
- Where does the data come from? Is it collected in-house, or sourced elsewhere?

---

### Decision · Program_Chairs · 2025-11-05

**Decision:**

Accept

**Comment:**

The abstract is of interest to the community and should be presented at the conference.